# Skeletal Muscle Discomfort and Lifestyle of Brazilian Military Police Officers of Administrative and Tactical Force

**DOI:** 10.3390/jfmk8040148

**Published:** 2023-10-25

**Authors:** Renan Ribeiro de Oliveira, Jadder Bento da Costa Aquino, Carlos H. O. Reis, Geanderson S. Oliveira, Leonardo A. Vieira, Alexandre F. Machado, Roberta L. Rica, Valentina Bullo, Marco Bergamin, Stefano Gobbo, Danilo S. Bocalini

**Affiliations:** 1Experimental Physiology and Biochemistry Laboratory, Center of Physical Education and Sport, Federal University of Espírito Santo, Av. Fernando Ferrari, 514-Goiabeiras, Vitoria 29075-910, ES, Brazil; renan.ro99@gmail.com (R.R.d.O.); jadderbento11@gmail.com (J.B.d.C.A.); carlos.h.reis@edu.ufes.br (C.H.O.R.); geanderson.sampaio@gmail.com (G.S.O.); lcaramuru@gmail.com (L.A.V.); xdmachado@gmail.com (A.F.M.); robertarica@hotmail.com (R.L.R.); danilo.bocalini@ufes.br (D.S.B.); 2Department of Physical Education, Center of Physical Education and Sport, Federal University of Espírito Santo, Av. Fernando Ferrari, 514-Goiabeiras, Vitoria 29075-910, ES, Brazil; 3Department of Medicine, University of Padova, 35131 Padova, PD, Italy; valentina.bullo@unipd.it (V.B.); stefano.gobbo@unipd.it (S.G.)

**Keywords:** public safety, muscular pain, police, military, physical activity

## Abstract

Our aim was to evaluate musculoskeletal discomfort and the lifestyle of military police officers of administrative and tactical force departments. Military police officers were distributed into two groups: administrative (Adm, *n* = 15) and tactical force (TF, *n* = 16) departments. Their lifestyle was assessed using the Fantastic Lifestyle questionnaire. Moreover, physical activity quantification was assessed using the International Physical Activity questionnaire, and musculoskeletal discomfort was quantified using the Corlett diagram. The mean total time of physical activity was 546 ± 276 min per week. No differences (*p* = 0.0832) were found between the Adm (454 ± 217 min) and TF (623 ± 301 min) groups. Concerning lifestyle, in general the sample presented very good (42%) and good (42%) style classification. For this parameter, no significant differences were found, but only a tendency was discovered (x^2^: 7.437; *p* = 0.0592); indeed, the TF presented a better classification (63%) of very good, compared to the Adm (53%) of good. No differences (*p* > 0.05) were found in musculoskeletal perception of discomfort between the right and left sides (*p* > 0.05) for all police officers and between the Adm and FT groups (*p* > 0.05). Military police officers showed high and moderate risk for waist circumference and waist-to-hip ratio, respectively; however, lifestyle and total time of physical activity were considered adequate without differences between military administrative and tactical force sectors.

## 1. Introduction

The military profession provides to its operating agents tasks of different physical demands on a daily basis, from long periods with less demanding activities (typing occurrences, driving a car), to critical situations, with short periods of physically demanding tasks such as running, crawling, jumping, lifting, pushing, pulling, and transporting objects or people, and controlling uncooperative suspects [1]. Araújo et al. stated that the effectiveness of these tasks is largely determined by the somatic characteristics of the police element that performs them, so a military officer whose body morphology is compromised is much more subject to being attacked and overcome [2]. In addition, these individuals live with stressful, dangerous, and demanding situations in their work shifts; at the same time that they are forced to use heavy clothing and personal protective equipment, increasing physical demands and psychological stressors.

Some military personnel, due to the characteristics of the work, may spend time carrying large amounts of weight (work equipment), but at other times may also remain frequently seated for a long time. This combination of mandatory equipment, such as the ballistic vest, tactical belt, leg holster, weapons with their main chargers and loaded spares, boots, handcuffs, radio communicator, and flashlight, and the large amount of time that military police remain seated, especially for motorized patrols, for example, can impact musculoskeletal health and physical fitness indicators that contribute to performance in operational tasks [3,4,5,6,7,8].

According to Calheiros et al., military police officers in the field, responsible for ostensive patrolling, are more predisposed to problems related to the spine, especially in the thoracic and lumbar regions, due to the maintenance of orthostatic postures for prolonged periods, and the aggravation of the ballistic vest [9]. Therefore, the professional activity of the police officer has characteristics that contribute to the onset of musculoskeletal disorders [10] and a natural predisposition to low back pain, due to the working day, time spent in a standing position, use of equipment, and physical and emotional stress [11].

Because of these well-known conditions, assessing the risk of incurring musculoskeletal disorders and injuries for these particular workers is essential to provide tailored interventions and occupational health protection measures. However, to the best of our knowledge, although the analysis of musculoskeletal discomfort is an intervention that has already been consolidated in clinical practice and ergonomics, information on military police is still inconclusive. Considering that operational military police officers are more likely to remain inactive due to motorized patrolling, in this study we hypothesized that skeletal muscle discomfort as well as lifestyle would have a greater impact in military from the tactical force. Additionally, the anthropometric, working time, and lifestyle parameters could be associated with skeletal muscle discomfort. Thus, the objective of this cross-sectional correlational study was to evaluate the musculoskeletal discomfort and the lifestyle of military police officers in the administrative and operational tactical force sectors.

## 2. Materials and Methods

### 2.1. Participants

After approval by the Research Ethics Committee of the Federal University of Espírito Santo (no. 6.275.609/2023), military police officers working in the 17th Independent Company of the Military Police of the municipality of Vila Velha, Espírito Santo, were invited to participate in the study. The invitation to participate in the study was carried out through direct contact between the researchers and the military and through verbal and digital dissemination strategies. Being active in employment was adopted as an inclusion criterion and recent return to work (3 months) as an exclusion criterion. Individuals who answered the questionnaire incorrectly and who did not present the signed informed consent form were excluded.

The company is composed of 120 military personnel: 20 from the administrative sector and 21 from the operational tactical force, with the rest having other functions. Of the 41 personnel from the administrative or operational tactical forces, only 10 subjects did not agree to participate in the study. Thus, the study sample comprised 31 military police officers, distributed into two groups: military from the administrative sector (Adm, *n* = 15) and military from the operational tactical force (TF, *n* = 16).

### 2.2. Evaluated Parameters

#### 2.2.1. Anthropometric Parameters

Height was measured using a Stadiometer, model WCS (Cardiomed, Brasilia, Brazil) with a precision of 0.1 cm. Body mass was measured using a Scale, Personal Line Model 150 (Filizola, São Paulo, Brazil) with an accuracy of 0.1 kg. The body mass index (BMI, kg/m^2^) was calculated according to the following equation: BMI = weight/height^2^. The circumferences of the abdomen, waist, and hips were evaluated using an anthropometric tape (Sanny^®^, São Bernardo do Campo, Brazil) to estimate abdominal adiposity and cardiovascular risk. The abdomen circumference was measured at the level of the umbilicus for men and women; the waist was measured at the midpoint between the last lower rib and the iliac crest for men and at the greatest protrusion of the buttocks for women; and, finally, the hips were measured at the largest circumference around the buttocks. Anthropometric variables were classified according to the recommendations of the World Health Organization [12].

#### 2.2.2. Musculoskeletal Discomfort

For the assessment of musculoskeletal discomfort, the diagram of Corlett and Manenica was used [13]. This instrument indicates the existence of pain, the painful area, and the intensity of the pain, by dividing the body into 27 regions and using a pain index that ranges between 1 (absence of pain) and 5 (extreme pain). Lifestyle assessment was performed using the “Fantastic Lifestyle” Questionnaire validated for the Brazilian population by Rodriguez-Añez et al. [14]. This questionnaire is a self-administered instrument that considers the behavior of individuals in the last month and allows the association between lifestyle and health to be determined based on 25 questions distributed into 9 domains: (1) family and friends; (2) physical activity; (3) nutrition; (4) cigarettes and drugs; (5) alcohol; (6) sleep, seat belt, stress, and safe sex; (7) type of behavior; (8) introspection; (9) work. The 25 questions that make up the body of the questionnaire were arranged on the Likert scale, so that 23 of these have five possible alternatives as an answer, and 2 are presented in a dichotomous manner. The following scores were used: excellent (85 to 100 points), very good (70 to 84 points), good (55 to 69 points), fair (35 to 54 points), and needs improvement (0 to 34 points).

#### 2.2.3. Physical Activity Level

The assessment of the military personnel’s physical activity level was performed using the International Physical Activity Questionnaire (IPAQ), short version [15,16,17]. Those who met the minimum recommendation of 150 min of weekly physical activity were classified as physically active (very active and active), and those who did not meet this recommendation were classified as inactive (inactive and insufficiently active), as established by the World Health Organization [18].

### 2.3. Statistical Analyses 

Data are presented as absolute frequency (F), relative (%) for qualitative variables, and mean and standard deviation for quantitative variables. The x^2^ test and the unpaired *t*-test were used to compare the data of the qualitative and quantitative variables, respectively. In addition, Pearson’s correlation was used to identify the correlation between length of service and perception of general pain with anthropometric parameters, level of physical activity, and lifestyle. The software GraphPad Prism version 6.00 for Windows (GraphPad Software, La Jolla, CA, USA) was used, adopting a significance level of *p* < 0.05.

## 3. Results

Of 31 participants, 4 (13%) were women working in the administrative sector, and 27 (87%) were men, of which 11 were from the Adm group and 16 were from the TF group. The general mean age of the participants was 32 ± 6 years, with 35 ± 6 years in the Adm group and 29 ± 3 years in the FT group, which was a significative difference (*p* = 0.0043).

Regarding operational function, in general the sample consisted of 20 (65%) soldiers, 5 (16%) corporals, 3 (10%) sergeants, 1 (3%) lieutenant, 1 (3%) captain and 1 (3%) major. The TF group was composed of 13 (81%) soldiers and 3 (19%) corporals. The Adm group comprised 7 (47%) soldiers, 2 (13.5%) corporals, 3 (20%) sergeants, 1 (6.5%) captain, 1 (6.5%) lieutenant, and 1 (6.5%) major. No differences were found between groups regarding operational function (x^2^: 7.946; *p* = 0.1576).

The general average service length corresponded to 9.04 ± 5.64 years of activity. Significant differences (*p* = 0.0014) were found between the service time of military personnel in the Adm (12.63 ± 5.65 years) and TF (6.38 ± 3.70 years) groups. Regarding the level of physical activity, military police officers in general were classified as active, with a mean total time of physical activity of 541 ± 272 (coefficient of variation 50.40%) minutes per week. No significant differences (*p* = 0.0832) were found in the time of weekly physical activity practice between the active military in the Adm (454 ± 217 min, coefficient of variation 47.73%) and TF (623 ± 301 min, coefficient of variation 48.26%) groups. The sample characteristics are shown in Table 1.

Although the general classification of BMI of 26.28 ± 4.07 kg/m^2^ was considered overweight, when comparing the soldiers of the Adm (eutrophic: 44%, overweight: 50%, obese I: 6%) and TF (eutrophic: 50%, overweight: 44%, obese I: 6%) groups, no significant differences were found (x^2^: 0.423; *p* = 0.8093). However, these results may be related to more lean mass instead of fat mass.

Regarding the risk classification for abdominal circumference (AC), the general average of the military personnel was normal, with 49% of the personnel having normal AC, 13% having medium risk, 32% having high risk, and 6% having very high risk. No significant difference was found (x^2^: 1.060; *p* = 0.7868) between the Adm group (normal: 47%, medium risk: 7%, high risk: 40%, very high risk: 6%) and the TF group (normal: 50%, medium risk: 19%, high risk: 25%, very high risk: 6%). Concerning waist circumference (WC), the military personnel in general were classified as low risk, with 35% presenting high risk and 3% very high risk. No significant differences (x^2^: 1.840; *p* = 0.3984) were revealed between soldiers in the Adm group (high risk: 27%, very high risk: 6%) and those in the TF group (56% for high risk). Regarding the risk classification according to the waist-to-hip ratio, in general, the military personnel were considered low risk; however, 29% presented low risk, 49% moderate risk, 16% high risk, and 6% very high risk. When comparing the ratings of the military, no differences were found in risk (x^2^: 2.837; *p* = 0.4174) between the Adm (low risk: 20%, moderate risk: 47%, high risk: 27% and very high risk) and TF (low risk: 38%, moderate risk: 50%, high risk: 6% and very high risk: 6%) groups.

As regards the assessment of lifestyle (Table 2), in general the sample presented a style classification of very good (42%) and good (42%). For this parameter, no significant differences were found, but only a tendency was discovered (x^2^: 7.437; *p* = 0.0592); indeed, the TF group presented a better classification (63%) of very good compared to the Adm (53%) group of good.

Table 3 and Table 4 show the results related to perception of musculoskeletal discomfort. No significant differences (*p* > 0.05) were found in the right and left sides between administrative and tactical force groups. Similarly, no differences (*p* > 0.05) were revealed in the perception of discomfort between the Adm and TF police officers among anatomical regions.

As shown in Table 5, no significant correlation was found between length of service and general pain, or between anthropometric parameters, lifestyle, and time of weekly physical activity.

## 4. Discussion

The aim of this study was to assess the lifestyle and musculoskeletal discomfort of military police officers in the administrative and operational sector of Espírito Santo district. No difference between administrative and operational officers was found.

Sample characteristics of administrative and tactical force showed no significant difference, except for age. Moreover, participants were classified as overweight but low risk for cardiovascular diseases, with no differences between military police officers in the administrative sector and ordinary troops. The high prevalence of overweight military police officers is not a new or recent finding [19,20,21,22]. If fact, most police officers are overweight, with a high abdominal and waist circumference, and this is associated with a higher risk of cardiovascular disease, diabetes, and musculoskeletal injuries [23,24]. However, these findings are in opposition to those from the study of Jorge et al., in which military police officers from the operational troop had a significantly higher BMI than police officers from the administrative sector [25]. It is probable that the lack of differences in the level of physical activity between the two groups of police officers may have determined the lack of differences in the BMI. The work activities carried out by military police officers in the administrative sector involve long periods of sedentary behavior, in a sitting posture and with exposure to screens, while the work routine of the operational troop involves shifts, night shifts, and overtime, which makes it difficult to adopt an active lifestyle. According to Bernardo et al., police officers who work in the administrative department have a probability of 0.927 of being less active than police officers who work in the operational department [26]. Thus, police officers in the administrative sector were expected to have a lower level of physical activity, which was not found in our study. However, there is evidence in the literature that demonstrates that the occupational activities of military police officers, in general, are predominantly related to sedentary behavior [17], and operational troop police officers had a higher level of physical activity compared to those in the administrative sector [17,25,26].

Regarding lifestyle, the military police were classified as good or very good, and better results were detected among operational troops compared to those in the administrative sector; however, no differences were revealed between the groups. The poorer lifestyle classification of police officers in the administrative sector may be explained by older age, longer service, greater body mass, and less time practicing physical activity [27,28]. Unpublished data of our laboratory demonstrated that military police officers from the metropolitan region of Vitória/ES, with a mean age, BMI, and length of service similar to those of the participants in our study, also had a good or very good lifestyle. In addition, Prando et al. pointed out that the lower prevalence of chronic diseases in military police officers in the metropolitan region of Vitória/ES was related to a healthy lifestyle [29], which could be related to socioeconomic and environmental characteristics [30,31,32].

Regarding musculoskeletal discomfort, we found that the perception of pain in military police officers can be considered low. In addition, no differences in pain perception were found between military police officers in the administrative sector and those in operational troops. However, the latter reported greater pain intensity in the dorsal region, particularly in the lumbar spine, which is still the most reported site of pain in both groups. Previous studies have also shown that low back pain is the main pain complaint in military police officers [19,33,34]. The systematic review carried out by Marins et al. demonstrated that low back pain is the most frequent musculoskeletal symptom in military police officers and has a prevalence between 42 and 52% [35]. According to the National Health Survey, the prevalence of low back pain in the Brazilian population in 2013 and 2019 was 18.5% and 21.6%, respectively [36,37]. The higher prevalence of low back pain in military police officers compared to the general population shows that specific preventive measures are needed for this professional category. The use of heavy equipment by military police, such as ballistic vests, is often associated with the manifestation of low back pain, mainly by operational troops [10,38,39]. In addition, sedentary behavior, especially sitting for long periods by police officers in the administrative sector, is also identified as a risk factor for the occurrence and worsening of low back pain symptoms [40,41,42]. Proper weight distribution of these devices on the body can contribute to prevention, and the use of leg holsters has been identified as one of the possible solutions for reducing overload on the lumbar spine [43]. Moreover, the practice of physical activity and exercise is recognized as an effective measure for the prevention and control of pain and improvement of physical function [44].

Additionally, some important limitations are present in this study, including the very small sample, which was limited to administrative and tactical force military police officers; low quantitative distribution of men and women in the sample; assessments of skeletal discomfort without identifying the primary cause; lack of objective parameters of skeletal muscle strength associated with skeletal discomfort; and evaluation of only the body composition and physical activity behavior, which limits the generalization of the results.

In conclusion, although the evaluated military police officers presented high and moderate risk considering waist circumference and waist-to-hip ratio, lifestyle and total time of physical activity were considered adequate and did not show indications of musculoskeletal discomfort. Nor were differences found between military personnel working in the administrative sector and tactical strength troops. New studies need to be carried out in order to identify the influence of personal, environmental, and socioeconomic factors on the lifestyle of military police officers from Espírito Santo, as well as the relationships between work characteristics, use of protective equipment, and length of service in the occurrence of musculoskeletal discomfort.

## Figures and Tables

**Table 1 jfmk-08-00148-t001:** Sample characteristics of administrative and tactical force police officers.

Parameters	Overall	Administrative	Tactical Force	*p*
Age (years)	32 ± 6	35 ± 6	29 ± 3	=0.0043
Body mass (kg)	81.93 ± 15.08	83.17 ± 19.06	80.70 ± 10.21	=0.6631
Height (m)	1.76 ± 0.08	1.76 ± 0.09	1.77 ± 0.07	=0.7699
BMI (kg/m^2^)	26.28 ± 4.07	26.70 ± 4.90	25.85 ± 3.16	=0.5774
AC (cm)	90.26 ± 10.27	91.63 ± 12.45	88.90 ± 7.71	=0.4781
WC (cm)	86.28 ± 9.71	87.77 ± 11.78	84.79 ± 7.21	=0.4106
HC (cm)	101.29 ± 8.24	102.35 ± 8.52	100.24 ± 8.09	=0.4934
WHR	0.85 ± 0.06	0.86 ± 0.07	0.85 ± 0.05	=0.7000

Values expressed as mean ± standard deviation. BMI: body mass index; AC: abdominal circumference. WC: waist circumference. HC: hip circumference. WHR: waist-to-hip ratio.

**Table 2 jfmk-08-00148-t002:** Lifestyle parameters of administrative and tactical force police officers.

Parameters	Overall	Administrative	Tactical Force	*p*
Family and friends	6.83 ± 1.82	6.67 ± 2.09	7.00 ± 1.56	=0.6250
Physical activity	5.70 ± 2.34	5.07 ± 2.71	6.33 ± 1.76	=0.1421
Nutrition	6.13 ± 2.65	6.60 ± 2.20	5.67 ± 3.04	=0.3442
Smoke and drugs	13.83 ± 1.60	14.20 ± 1.21	13.47 ± 1.88	=0.2167
Alcohol	9.97 ± 2.92	10.60 ± 2.23	9.33 ± 3.44	=0.2428
Sleep	13.37 ± 3.64	13.67 ± 4.53	13.07 ± 2.60	=0.6608
Behavior	4.33 ± 1.94	4.60 ± 2.50	4.07 ± 1.16	=0.4628
Introspection	8.63 ± 2.45	8.07 ± 2.71	9.20 ± 2.43	=0.2379
Work	3.23 ± 1.01	3.33 ± 0.82	3.13 ± 1.19	=0.5957
Total score	69.71 ± 16.42	72.80 ± 11.12	66.81 ± 20.12	=0.3118
Classification				
Excellent	3 (10%)	3 (10%)	0 (0%)	=0.0592
Very good	13 (42%)	3 (20%)	10 (63%)	
Good	13 (42%)	8 (53%)	5 (31%)	
Regular	2 (6%)	1 (7%)	1 (6%)	
Need better	-	-	-	

Values expressed as mean ± standard deviation.

**Table 3 jfmk-08-00148-t003:** Perception of musculoskeletal discomfort of administrative and tactical force police officers.

Parameters		Overall	Administrative	Tactical Force	*p*
Neck		1.79 ± 1.16	1.70 ± 1.21	1.88 ± 1.11	=0.6826
Upper back		1.61 ± 1.17	1.33 ± 1.05	1.88 ± 1.26	=0.2019
Middle back		1.63 ± 1.15	1.47 ± 1.06	1.78 ± 1.25	=0.4553
Lower back		2.10 ± 1.25	2.10 ± 1.24	2.09 ± 1.29	=0.9891
Pelvic		1.48 ± 1.18	1.27 ± 1.03	1.69 ± 1.30	=0.3258
Shoulder	Right side	2.00 ± 1.48	1.67 ± 1.29	2.31 ± 1.62	=0.2286
	Left side	1.97 ± 1.28	2.07 ± 1.22	1.88 ± 1.36	=0.6826
Arm	Right side	1.26 ± 0.89	1.67 ± 1.29	1.44 ± 1.21	=0.2480
	Left side	1.03 ± 0.18	1.07 ± 0.26	1.00 ± 0.00	=0.3343
Forearm	Right side	1.19 ± 0.75	1.07 ± 0.26	1.31 ± 1.01	=0.3616
	Left side	1.06 ± 0.25	1.07 ± 0.26	1.06 ± 1.25	=0.9639
Fist	Right side	1.45 ± 1.12	1.27 ± 0.70	1.63 ± 1.41	=0.3754
	Left side	1.32 ± 0.91	1.27 ± 0.70	1.38 ± 1.09	=0.7431
Hang	Right side	1.19 ± 0.79	1.13 ± 0.52	1.25 ± 1.00	=0.6844
	Left side	1.06 ± 0.36	1.00 ± 0.00	1.13 ± 0.50	=0.3332
Thig	Right side	1.26 ± 0.82	1.07 ± 0. 26	1.44 ± 1.09	=0.2053
	Left side	1.10 ± 0.30	1.07 ± 0. 26	1.13 ± 0.34	=0.5945
Leg	Right side	1.77 ± 1.33	1.53 ± 1.25	2.00 ± 1.41	=0.3370
	Left side	1.65 ± 1.17	1.60 ± 1.24	1.69 ± 1.44	=0.8308
Ankle and feet	Right side	1.58 ± 1.26	1.27 ± 1.01	1.88 ± 1.41	=0.1795
	Left side	1.71 ± 1.30	1.47 ± 1.13	1.94 ± 1.44	=0.3167
General discomfort		31.23 ± 13.61	28.53 ± 11.25	33.75 ± 15.45	=0.2897

Values expressed as mean ± standard deviation.

**Table 4 jfmk-08-00148-t004:** Perception of skeletal muscle discomfort classification of administrative and tactical force police officers.

Parameters		Overall					Administrative					Tactical Force				
		Skeletal Muscle Discomfort Classification					Skeletal Muscle Discomfort Classification					Skeletal Muscle Discomfort Classification				
		A F (%)	L F (%)	M F (%)	I F (%)	E F (%)	A F (%)	L F (%)	M F (%)	I F (%)	E F (%)	A F (%)	L F (%)	M F (%)	I F (%)	E F (%)
Neck		19 (61)	6 (19)	3 (10)	1 (3)	2 (6)	11 (73)	2 (13)	0 (0)	1 (7)	1 (7)	8 (50)	4 (25)	3 (19)	0	1 (6)
Upper back		23 (74)	3 (10)	2 (6)	1 (3)	2 (6)	14 (93)	0 (0)	0 (0)	0 (0)	1 (7)	9 (56)	3 (19)	2 (13)	1 (6)	1 (6)
Middle back		22 (71)	5 (16)	1 (3)	1 (3)	2 (6)	11 (73)	3 (20)	0 (0)	0 (0)	1 (7)	11 (69)	2 (13)	1 (6)	1 (6)	1 (6)
Lower back		17 (55)	5 (16)	5 (16)	2 (6)	2 (6)	8 (53)	2 (13)	2 (13)	2 (13)	1 (7)	9 (56)	3 (19)	3 (19)	0 (0)	1 (6)
Pelvic		27 (87)	1 (3)	1 (3)	0 (0)	2 (6)	14 (93)	0 (0)	0 (0)	0 (0)	1 (7)	13 (81)	1 (6)	1 (6)	0 (0)	1 (6)
Shoulder	Right side	19 (61)	3 (10)	3 (10)	2 (6)	4 (13)	11 (73)	1 (7)	1 (7)	1 (7)	1 (7)	8 (50)	2 (13)	2 (13)	1 (6)	3
	Left side	17 (55)	4 (13)	6 (19)	2 (6)	2 (6)	7 (47)	2 (13)	5 (33)	0 (0)	1 (7)	10 (63)	2 (13)	1 (6)	2 (13)	1 (6)
Arm	Right side	28 (90)	1 (3)	0 (0)	1 (3)	1 (3)	14 (93)	1 (7)	0 (0)	0 (0)	0 (0)	14 (88)	0 (0)	0 (0)	1 (6)	1 (6)
	Left side	30 (97)	1 (3)	0 (0)	0 (0)	0 (0)	14 (93)	1 (7)	0 (0)	0 (0)	0 (0)	16 (100)	0 (0)	0 (0)	0 (0)	0
Forearm	Right side	28 (90)	2 (6)	0 (0)	0 (0)	1 (3)	14 (93)	1 (7)	0 (0)	0 (0)	0 (0)	14 (88)	1 (6)	0 (0)	0 (0)	1 (6)
	Left side	29 (94)	2 (6)	0 (0)	0 (0)	0 (0)	14 (93)	1 (7)	0 (0)	0 (0)	0 (0)	15 (94)	1 (6)	0 (0)	0 (0)	0 (0)
Fist	Right side	26 (84)	0 (0)	3 (10)	0 (0)	2 (6)	13 (87)	0 (0)	2 (13)	0 (0)	0 (0)	13 (81)	0 (0)	1 (6)	0 (0)	2 (13)
	Left side	27 (87)	0 (0)	3 (10)	0 (0)	1 (3)	13 (87)	0 (0)	2 (13)	0 (0)	0 (0)	14 (88)	0 (0)	1 (6)	0 (0)	1 (6)
Hang	Right side	29 (94)	0 (0)	1 (10)	0 (0)	1 (3)	14 (93)	0 (0)	1 (7)	0 (0)	0 (0)	15 (94)	0 (0)	0 (0)	0 (0)	1 (6)
	Left side	30 (97)	0 (0)	1 (10)	0 (0)	0 (0)	15 (100)	0 (0)	0 (0)	0 (0)	0 (0)	15 (94)	0 (0)	1 (6)	0 (0)	0 (0)
Thig	Right side	27 (87)	2 (6)	1 (3)	0 (0)	1 (3)	14 (93)	1 (7)	0 (0)	0 (0)	0 (0)	13 (81)	1 (6)	1 (6)	0 (0)	1 (6)
	Left side	28 (90)	3 (10)	0 (0)	0 (0)	0 (0)	14 (93)	1 (7)	0 (0)	0 (0)	0 (0)	14 (88)	2 (13)	0 (0)	0 (0)	0 (0)
Leg	Right side	22 (71)	3 (10)	3 (10)	1 (3)	2 (6)	12 (80)	1 (7)	0 (0)	1 (7)	1 (7)	10 (63)	2 (13)	3 (19)	0 (0)	1 (6)
	Left side	21 (68)	5 (16)	2 (6)	1 (3)	2 (6)	11 (73)	2 (13)	0 (0)	1 (7)	1 (7)	10 (63)	3 (19)	2 (13)	0 (0)	1 (6)
Ankle and feet	Right side	24 77)	2 (6)	2 (6)	0 (0)	3 (10)	14 (93)	0 (0)	0 (0)	0 (0)	1 (7)	10 (63)	2 (13)	2 (13)	0 (0)	2 (13)
	Left side	22 (71)	2 (6)	4 (13)	0 (0)	3 (10)	12 (80)	1 (7)	1 (7)	0 (0)	1 (7)	10 (63)	1 (6)	3 (19)	0 (0)	2 (13)

Skeletal muscle discomfort classification: absence of pain/discomfort (A), low pain/discomfort (L), moderate pain/discomfort (M), intense pain/discomfort (I), and extreme pain/discomfort (E).

**Table 5 jfmk-08-00148-t005:** Correlation of anthropometric parameters and lifestyle, service time, and pain perception of police officers.

Parameters	Service Time			Pain Perception		
	r	*p*	95% CI	r	*p*	95% CI
Service time	-	-	-	-	-	-
Body mass	−0.284	=0.1278	−0.4549–0.2575	−0.268	=0.1520	−0.5730–0.1020
BMI	−0.178	=0.3459	−0.5060–0.1945	−0.182	=0.3339	−0.5094–0.1901
AC	−0.142	=0.4530	−0.4781–0.2297	−0.129	=0.4951	−0.4680–0.2420
WC	−0.113	=0.5513	−0.4549–0.2575	−0.285	=0.1268	−0.5852–0.0838
HC	−0.186	=0.3231	−0.5125–0.1860	−0.185	=0.3276	−0.5112–0.1878
WHR	0.034	=0.8584	−0.3303–0.3895	−0.239	=0.2034	−0.5517–0.1327
Lifestyle	−0.110	=0.5616	−0.5385–0.1512	0.143	=0.4508	−0.2290–0.4786
Physical activity time	0.010	=0.9540	−0.4526–0.2603	0.047	=0.8010	−0.3179–0.4012
Pain perception	−0.221	=0.2403	−0.5385–0.1512	-	-	-

BMI: body mass index. AC: abdominal circumference. WC: waist circumference. HC: hip circumference. WHR: waist-to-hip ratio. 95% CI: 95% confidence intervals.

## Data Availability

Data presented in the current paper are available upon request.

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
