# Peer review of "Skeletal Muscle Discomfort and Lifestyle of Brazilian Military Police Officers of Administrative and Tactical Force"

_jfmk, 2023, doi:10.3390/jfmk8040148_

Round 1

Reviewer 1 Report

Comments and Suggestions for Authors

Manuscript ID: jfmk-2640750

Manuscript title: Skeletal muscle discomfort and lifestyle of Brazilian military police officers of administrative and tactical force

Comments

This manuscript reports a study designed to evaluate the musculoskeletal discomfort and the lifestyle of military police officers in the administrative and operational sectors.

Major comments

1. Introduction. I missed a formal statement of the study’s hypotheses. Please elaborate on them considering the statistical analysis you conducted.

2. Methods. I missed a formal description of the study design. I assume it is a cross-sectional correlational study.

3. Methods. Alongside the study design description, it is important to adhere to reporting guidelines to ensure all relevant information is disclosed. Please find the STROBE guidelines in the EQUATOR Network (https://www.equator-network.org/reporting-guidelines/strobe/) and report/structure your manuscript accordingly.

4. Results, tables 3a and 3b. I found no p-values related to the comparison of right vs. left body sides as stated in lines 165-166.

5. Results, table 4. Consider adding 95% confidence intervals for interpretation of the variability of the effect size. Even though the a priori significance level was not crossed by any p-value, it did not mean the effect was absent (it was just highly uncertain given your sample size).

6. Discussion. A discussion about the study's strengths and limitations is required. List the threats to your study's internal validity and how they can limit the generalization of your findings. Also, consider focusing on the interpretation of the strength of the observed effect sizes and how your results can help future research on the topic.

Minor comments

1. Lines 110, 159. Please use the correct symbol for the chi-square test.

2. Line 139. BMI classification works well for the general population. It is possible that the overweight classification is related to more lean mass instead of fat mass.

3. Line 178. As no correlation coefficient equals 0, I believe you mean you ‘observed no significant correlation’.

Author Response

We thank the Editor in Chief and Referees for the possibility to revise our manuscript. We tried to fulfill all the points indicated by the Referees. All adjustment can be found in the text (in red).

REVIEWER 1

Manuscript ID: jfmk-2640750

Manuscript title: Skeletal muscle discomfort and lifestyle of Brazilian military police officers of administrative and tactical force

Comments

This manuscript reports a study designed to evaluate the musculoskeletal discomfort and the lifestyle of military police officers in the administrative and operational sectors.

Major comments

  1. Introduction. I missed a formal statement of the study’s hypotheses. Please elaborate on them considering the statistical analysis you conducted.

AUTHORS: Thanks Reviewer#1 for the suggestion. We consider that operational military police officers are more likely to remain inactive due to motorized patrolling; in this study we hypothesized that skeletal muscle discomfort would be greater in military from the tactical force. Moreover, we hypothesized that anthropometric data, working time and life style parameters could be associated with skeletal muscle discomfort. 

  1. Methods. I missed a formal description of the study design. I assume it is a cross-sectional correlational study.

AUTHORS: Thanks Reviewer#1 for the suggestion. Yes, our study is a cross-sectional correlational study design. We insert this information in the paper.

  1. Methods. Alongside the study design description, it is important to adhere to reporting guidelines to ensure all relevant information is disclosed. Please find the STROBE guidelines in the EQUATOR Network (https://www.equator-network.org/reporting-guidelines/strobe/) and report/structure your manuscript accordingly.

AUTHORS: We used the strobe statement checklist to check the quality of our manuscript. Thanks Reviewer#1 for the methodology suggestion that we will use also for future.

  1. Results, tables 3a and 3b. I found no p-values related to the comparison of right vs. left body sides as stated in lines 165-166.

AUTHORS: Thanks Reviewer#1, you have reason. We add a new sentence correcting the misinterpretation.

  1. Results, table 4. Consider adding 95% confidence intervals for interpretation of the variability of the effect size. Even though the a priori significance level was not crossed by any p-value, it did not mean the effect was absent (it was just highly uncertain given your sample size).

AUTHORS: Thanks Reviewer#1. We add the 95% confidence intervals.

  1. Discussion. A discussion about the study's strengths and limitations is required. List the threats to your study's internal validity and how they can limit the generalization of your findings. Also, consider focusing on the interpretation of the strength of the observed effect sizes and how your results can help future research on the topic.

AUTHORS: Thanks Reviewer#1. We add new sentence regarding limitation.

Minor comments

  1. Lines 110, 159. Please use the correct symbol for the chi-square test.

AUTHORS: Thanks Reviewer#1 for the suggestion. We correct. 

  1. Line 139. BMI classification works well for the general population. It is possible that the overweight classification is related to more lean mass instead of fat mass.

AUTHORS: Thanks Reviewer#1 for the suggestion. We correct. 

  1. Line 178. As no correlation coefficient equals 0, I believe you mean you ‘observed no significant correlation’.

AUTHORS: Thanks Reviewer#1 for the suggestion. We correct. 

Reviewer 2 Report

Comments and Suggestions for Authors

This is an interesting epidemiological study, however performed in a very small sample, and that is a problem for this type of research.

What was the initial aim of this study? To perhaps identify differences between the 2 samples compared in order to make suggestions on improvement of possible deficits identified?

Do you think the sample of administrative and tactical force officers used represented the population it referred to? I would have expected these two samples to be substantially different, but apparently not for the sample you selected.

From what is gathered from Tables 1 and 3b. the sample consisted of relatively young and mostly pain-free to low-pain participants. This significantly limits the clinical interpretation of your findings.

Having many overweight in the sample, as in other studies, is probably one of the more significant findings of this study.

Line 58: “well-known” instead of well-knowing

Line 60: “To” the best of our knowledge 

Comments on the Quality of English Language

No significant issues were identified.

Author Response

We thank the Editor in Chief and Referees for the possibility to revise our manuscript. We tried to fulfill all the points indicated by the Referees. All adjustment can be found in the text (in red).

REVIEWER 2

1. This is an interesting epidemiological study, however performed in a very small sample, and that is a problem for this type of research.

AUTHORS: Thanks Reviewer#2 for the suggestion. We add new sentence regarding limitation.

2. What was the initial aim of this study? To perhaps identify differences between the 2 samples compared in order to make suggestions on improvement of possible deficits identified?

AUTHORS: Thanks Reviewer#2 for the suggestion. We add a new sentence on the introduction section to clarify the hypothesis.

3. Do you think the sample of administrative and tactical force officers used represented the population it referred to? I would have expected these two samples to be substantially different, but apparently not for the sample you selected

AUTHORS: Thanks Reviewer#2 for the suggestion. This is a very interesting concern, and was negated according to our hypothesis in the introduction section.

4. From what is gathered from Tables 1 and 3b. the sample consisted of relatively young and mostly pain-free to low-pain participants. This significantly limits the clinical interpretation of your findings.

AUTHORS: Thanks Reviewer#2 for the suggestion. Our results independently from the sample age are different from data of other studies conducting on militaries officers (REF). However, our data are similar to results from studies conduct in Espirito Santo region. In this way, it’s possible to indicate that our results, could be influenced by several parameters, such as lifestyle of population from Espirito Santo.

5. Having many overweight in the sample, as in other studies, is probably one of the more significant findings of this study. 

AUTHORS: Thanks Reviewer#2 for the suggestion. It’s a very interesting point of view, however, its necessary to consider that we do not evaluate fat percentage. In this way is possible to consider that the high amount of overweight could be related to more lean mass instead of fat mass, but it remains an assumption.

6. Line 58: “well-known” instead of well-knowing

AUTHORS: Thanks Reviewer#2 for the suggestion. We correct.

7. Line 60: “To” the best of our knowledge 

AUTHORS: Thanks Reviewer#2 for the suggestion. We correct.

Reviewer 3 Report

Comments and Suggestions for Authors

The authors have compared a series of lifestyle and cardiovascular risk variables between two groups of military police.

Material and method.

Authors should indicate precisely where the trunk circumferences are measured and the type of tape measure used.

Results:

Reduce the decimal places of the ages to one.

The authors could provide information on the coefficient of variation of the data they provide in the paragraph of lines 130-134, to establish their dispersion. Possibly the absence of significant difference could be due to the fact that the groups are not very homogeneous having high intragroup variability. If this is so, you can comment on it in the discussion.

The results paragraph, of lines 143-161, is very cumbersome, it could be replaced by a table. The values of the population as a whole are of no interest. The interesting thing is the comparisons.

The point on line 157 (.... 6%). As regards ....) should be to separate a new paragraph dedicated to "lifestyle".

Table 2 the values of the population as a whole (overall) are of no interest.

Table 3a Why is the discomfort of the population as a whole higher than that of each group separately? The authors should review the data, there may be a typo in the points of "general discomfort" there are many decimals.

Table 3b. Including authors in the general population column makes data comparison more difficult. As that column is the sum of the cases of the next two, it does not contribute anything. Authors must assess whether to maintain or eliminate it. For me, as a reader, it has been difficult for me to interpret the data.

I think that making assessments of pain, without identifying the cause, does not make sense. Combining in the same group the pain due to a contusion, which will disappear in a few days, with the pain of an osteoarthritis that will remain over time is not representative. Authors should devote a comment to these as "limitations" of their work.

It is necessary for authors to meditate and write about the limitations of their study. It is also interesting that they include a paragraph, in the discussion, on the applicability of their work and possible recommendations to improve the standard of living of their population.

 Authors should review references. There are quite a few mistakes.

Author Response

We thank the Editor in Chief and Referees for the possibility to revise our manuscript. We tried to fulfill all the points indicated by the Referees. All adjustment can be found in the text (in red).

REVIEWER 3

The authors have compared a series of lifestyle and cardiovascular risk variables between two groups of military police.

Material and method.

1. Authors should indicate precisely where the trunk circumferences are measured and the type of tape measure used.

AUTHORS: Thanks Reviewer#3 for the suggestion. We correct.

Results:

2. Reduce the decimal places of the ages to one.

AUTHORS: Thanks Reviewer#3 for the suggestion. We correct.

3. The authors could provide information on the coefficient of variation of the data they provide in the paragraph of lines 130-134, to establish their dispersion. Possibly the absence of significant difference could be due to the fact that the groups are not very homogeneous having high intragroup variability. If this is so, you can comment on it in the discussion.

AUTHORS: Thanks Reviewer#3 for the suggestion. According to the referee we add coefficient of variation.

4. The results paragraph, of lines 143-161, is very cumbersome, it could be replaced by a table. The values of the population as a whole are of no interest. The interesting thing is the comparisons.

AUTHORS: Thanks Reviewer#3 for the suggestion. We agree with the referee point of view. However, for us the values of overall sample are important to give the readers a view of the entire police officer populations. These data could be used in future studies for a comparison.

5. The point on line 157 (.... 6%). As regards ....) should be to separate a new paragraph dedicated to "lifestyle".

AUTHORS: Thanks Reviewer#3 for the suggestion. We change as indicated.

6. Table 2 the values of the population as a whole (overall) are of no interest.

AUTHORS: Thanks Reviewer#3 for the suggestion. We reply in the point 4.

7. Table 3a Why is the discomfort of the population as a whole higher than that of each group separately? The authors should review the data, there may be a typo in the points of "general discomfort" there are many decimals.

AUTHORS: Thanks Reviewer#3 for the suggestion. We change as indicated.

8. Table 3b. Including authors in the general population column makes data comparison more difficult. As that column is the sum of the cases of the next two, it does not contribute anything. Authors must assess whether to maintain or eliminate it. For me, as a reader, it has been difficult for me to interpret the data.

AUTHORS: Thanks Reviewer#3 for the suggestion. We reply in the point 4.

9. I think that making assessments of pain, without identifying the cause, does not make sense. Combining in the same group the pain due to a contusion, which will disappear in a few days, with the pain of an osteoarthritis that will remain over time is not representative. Authors should devote a comment to these as "limitations" of their work.

AUTHORS: Thanks Reviewer#3 for the suggestion. We add new sentence regarding limitation in the discussion section.

10. It is necessary for authors to meditate and write about the limitations of their study. It is also interesting that they include a paragraph, in the discussion, on the applicability of their work and possible recommendations to improve the standard of living of their population.

AUTHORS: Thanks Reviewer#3 for the suggestion. We add new sentence regarding limitation in the discussion section.

11. Authors should review references. There are quite a few mistakes.

AUTHORS: Thanks Reviewer#3 for the suggestion. We revised.

Round 2

Reviewer 1 Report

Comments and Suggestions for Authors

Thank you for the opportunity to discuss your manuscript. All comments were adequately addressed.

Author Response

Reviewer 1

Comments and Suggestions for Authors

Thank you for the opportunity to discuss your manuscript. All comments were adequately addressed.

AUTHORS: Thanks Reviewer#1 for the suggestions and the time spent.

Reviewer 2 Report

Comments and Suggestions for Authors

I believe Table 4 is redundant. Otherwise, the manuscript is now improved, and can proceed to publication.

Comments on the Quality of English Language

Minor editing required.

Author Response

Reviewer 2

I believe Table 4 is redundant. Otherwise, the manuscript is now improved, and can proceed to publication.

AUTHORS: Thanks Reviewer#2 for the suggestions and the time spent.

Reviewer 3 Report

Comments and Suggestions for Authors

The authors have clarified and completed my suggestions from the first round.

Despite this, there are still errors in the references. For example: there is an excess of capital letters in references 11, 15 and 17.

Author Response

Reviewer 3

The authors have clarified and completed my suggestions from the first round.

Despite this, there are still errors in the references. For example: there is an excess of capital letters in references 11, 15 and 17.

AUTHORS: Thanks Reviewer#3 for the suggestions and the time spent. We made amendments in text and reference sections.